# Safe Zones for Facial Fillers: Anatomical Study of SubSMAS Spaces in Asians

**DOI:** 10.3390/diagnostics14131452

**Published:** 2024-07-08

**Authors:** Gi-Woong Hong, Hyewon Hu, Youngjin Park, Hyun Jin Park, Kyu-Ho Yi

**Affiliations:** 1Sam Skin Plastic Surgery Clinic, Seoul 06577, Republic of Korea; cosmetic21@hanmail.net; 2Division in Anatomy and Developmental Biology, Department of Oral Biology, Human Identification Research Institute, BK21 FOUR Project, Yonsei University College of Dentistry, 50-1 Yonsei-ro, Seodaemun-gu, Seoul 03722, Republic of Korea; wonhuh@yuhs.ac; 3Obliv Clinic, Incheon 21998, Republic of Korea; youngjinp@gmail.com; 4Department of Anatomy, Daegu Catholic University School of Medicine, Daegu 42472, Republic of Korea; 5Maylin Clinic (Apgujeong), Seoul 06005, Republic of Korea

**Keywords:** subSMAS spaces, filler procedures, facial anatomy, cadaver study, facial aesthetics

## Abstract

The study “Spaces of the Face for Filler Procedures: Identification of subSMAS Spaces Based on Anatomical Study” explores the anatomy of facial spaces crucial for safe and effective filler injections. By delineating the subSMAS (sub-superficial musculoaponeurotic system) spaces, this research highlights how these virtual compartments, bordered by fat, muscles, fascia, and ligaments, facilitate independent muscle movement and reduce the risk of damaging critical structures. The thicker and more robust skin of East Asians necessitates deeper filler injections, emphasizing the significance of accurately identifying these spaces. A cadaver study with dyed gelatin validated the existence and characteristics of these subSMAS spaces, confirming their safety for filler procedures. Key spaces, such as the subgalea-frontalis, interfascial and temporalis, and prezygomatic spaces, were examined, illustrating safe zones for injections. The findings underscore the importance of anatomical knowledge for enhancing facial aesthetics while minimizing complications. This study serves as a guide for clinicians to perform precise and safe filler injections, providing a foundation for further research on the dynamic interactions of these spaces and long-term outcomes.

## 1. Introduction

The facial spaces for filler procedures are virtual spaces that are enclosed and demarcated by fat compartments, muscles, fascia, and ligaments. The purpose of distinguishing these spaces is based on their significance. First, the spaces located between the deep fascia and the superficial fascia allow the facial muscles within each space to move independently without being impeded by or impeding other muscle layers [1,2,3,4]. For example, when the orbicularis oculi muscle or the orbicularis oris muscle in the superficial layer contracts and moves the areas around the eyes or mouth, the deeper zygomaticus major and minor muscles can function to lift the corners of the mouth without being affected by the movements of the superficial muscles [5,6,7]. Additionally, since blood vessels and nerves mainly pass through the boundaries of these facial spaces, the interiors of these spaces are relatively safe. Therefore, understanding these spaces well can help perform filler procedures more safely and easily [8,9].

Typically, soft consistent fillers used to improve facial wrinkles and volume loss can be injected into the subcutaneous fat layer beneath the skin, but firm consistent fillers require injection into deeper layers to avoid damaging the blood vessels and nerves, thus targeting areas below the SMAS (superficial musculoaponeurotic system) layer. Moreover, the skin of East Asians is thicker and heavier compared to that of Westerners, with more robust and tough structures such as the SMAS and retinacular cutis connecting the skin and subcutaneous tissue [5,10,11,12,13]. Therefore, to achieve sufficient facial volume, it is not efficient to inject fillers only into the subcutaneous fat layer below the skin. Hence, it is important to accurately identify the spaces beneath the SMAS layer and enhance the volume of these spaces.

As mentioned earlier, each area of the face is bordered by major ligamentous structures, and several spaces composed only of soft tissues such as fat are safe for filler procedures. Anatomically, these spaces can be referred to as safe subSMAS adipofascial spaces [14,15,16,17]. The boundaries and indications for filler procedures in these spaces are summarized in Table 1. To anatomically verify the subSMAS adipofascial spaces of the face, the authors conducted a cadaver study using dyed gelatin. Entry points were marked on various parts of the cadaver’s face, as occurs in filler procedures using cannulas. Small amounts of dyed gelatin were then injected into each subSMAS space through these entry points. After the gelatin had set, the face was carefully dissected layer by layer to confirm that the dyed gelatin had accurately filled the intended subSMAS adipofascial spaces [1,3,4,18].

It is essential to also examine the types of fascia and muscles forming the SMAS layer across the entire face, as well as the thickness of the midfacial SMAS layer. The midfacial SMAS layer does not have a uniform thickness across the face; it is typically thickest in the preauricular region and gradually thins toward the medial facial region. During actual cadaver dissections, a distinct and tough SMAS layer resembling a white membrane could be observed on the side of the face, but in the central face, there was no white membrane-like appearance, making it look like only a fat layer existed (Figure 1). Previously, one explanation for the nasolabial fold was that a thick and tough SMAS layer existed outside the fold, but not inside it, creating a boundary. However, histological studies have shown that the SMAS extends even inside the fold, although it is thinner than outside, indicating the presence of a membrane surrounding the fat layer [19,20,21].

Upon the removal of the skin from the cadaver after the gelatin had set, the superficial fat could be seen, as shown in Figure 2A, but the gelatin injected into the subSMAS space was not visible. When the superficial fat layer was lifted, the SMAS layer was exposed, and the green gelatin injected beneath the SMAS layer became partially visible (Figure 2B). A more detailed examination of each subSMAS space in the face is provided in the following sections.

The space for fillers in the deep layer has not been discovered yet, and its differences based on age, sex, and ethnicity are not known. However, the objective of this study is to identify and delineate the subSMAS (sub-superficial musculoaponeurotic system) spaces in the face through detailed anatomical analysis. By understanding these spaces, the study aims to provide guidance for safe and effective filler injections, minimizing the risk of complications and enhancing facial aesthetics. The research focuses on the anatomical features that facilitate independent muscle movement and reduce the risk of damaging critical structures during filler procedures.

## 2. Materials and Methods

In this study, we used 10 fresh Korean cadavers to reveal the anatomical spaces relevant for filler injection. The cadavers were obtained from an institute that ensures all donations are approved for research purposes.

Prior to dissection, we injected color-dyed filler into specific areas of the face to visualize the subSMAS (sub-superficial musculoaponeurotic system) spaces. Entry points for the injections were carefully marked on each cadaver, mimicking the typical entry points used in clinical filler procedures. The filler was injected using cannulas to accurately target the subSMAS spaces.

Following the injection, the cadavers were dissected layer by layer. The dissection process involved careful removal of the skin and superficial fat to expose the SMAS layer. Further dissection revealed the subSMAS spaces filled with the dyed filler, allowing us to confirm their location and boundaries. Each space was documented and analyzed to provide a detailed map for safe and effective filler injections.

All procedures were conducted in compliance with ethical guidelines. The use of color-dyed filler and detailed dissection provided a clear visualization of the subSMAS spaces, contributing valuable insights for clinical applications in facial aesthetics.

## 3. Results

### 3.1. Subgalea-Frontalis Space

Beneath the galea-frontalis muscle, which corresponds to the SMAS in the forehead area, lies the subgalea-frontalis space. This space contains structures such as the superficial temporal artery and the temporal branch of the facial nerve entering from the outside of the forehead, as well as the supraorbital artery and nerve and the supratrochlear artery and nerve running vertically from the bottom to the top in the central forehead area [22,23,24,25,26,27]. When this area is lifted, the supraorbital and supratrochlear arteries and nerves emerging from the supraorbital and supratrochlear foramen or notch around the orbital rim can be identified. Therefore, the adipofascial space into which the dyed gelatin was injected is confirmed to be a space that does not damage important blood vessels and nerves (Figure 3).

### 3.2. Interfascial and Temporalis Space

In the temple area, if there is no severe hollowing, fillers are often injected into the subcutaneous layer below the skin, but since this area is very thin, it may appear uneven after the procedure. Therefore, it is now common to inject fillers between the superficial temporal fascia and the deep temporal fascia, which correspond to the SMAS. The temporal area has an upper temporal space bounded by extensions of the temporal ligament adhesion known as the STS (Superior Temporal Septum) and the ITS (Inferior Temporal Septum) and a lower temporal space bounded by the ITS and the zygomatic arch (Figure 4) [21,28]. The upper and lower temporal spaces are spaces formed between the superficial temporal fascia, which corresponds to the midfacial SMAS, and the deep temporal fascia. The upper temporal space does not contain other important structures, but the lower temporal space is an anatomically important triangular-shaped area containing structures such as the superficial temporal artery, the temporal branch of the facial nerve, and the medial and lateral branches of the zygomaticotemporal nerve. Moreover, the sentinel vein passes vertically through the muscle and fascia, running parallel to the middle temporal vein above the zygomatic arch, requiring caution during procedures [29,30]. The most important blood vessel and nerve in the temporal area, the superficial temporal artery and the temporal branch of the facial nerve, typically run within or just below the superficial temporal fascia. Therefore, maintaining the plane between the STF and DTF during the procedure can avoid damage to these structures. It is advisable to use a cannula for filler procedures in this space, with the entry point typically positioned at the junction of the zygomatic arch and the lateral orbital rim to avoid damage to structures while easily filling the hollowed space (Figure 5) [31,32,33].

When there is severe hollowing that requires a lot of volume enhancement, it is often assumed that injecting deeply beneath the temporalis muscle is the best approach. However, for effective volumization in the muscle’s deeper layer, the elasticity of the filler must be significantly high, and a large amount of filler is required. Another issue is the potential movement of the injected filler due to the strong contractions of the temporalis muscle during chewing, leading to changes in the physical properties of the filler over time, making the actual effect less efficient relative to the hardness and amount of injected filler. Patients with severe temporal hollowing often have more pronounced hollowing below the temporal arch rather than the upper temporal area. In such cases, filler can be injected into the superficial temporal fat pad located between the superficial layer and the deep layer of the deep temporal fascia, efficiently enhancing the volume above the zygomatic arch with a small amount of filler.

Technically, many practitioners find it difficult to accurately inject into this layer, but with experience, it is possible to feel the loose superficial temporal fat pad by penetrating the superficial temporal fascia and then the deeper, firmer superficial layer of the deep temporal fascia with a cannula. If the cannula is directed further downwards, it will be blocked by the very firm deep layer of the deep temporal fascia, naturally guiding the cannula to the correct position [19,34]. For patients with significant hollowing above the zygomatic arch, understanding the anatomical layers of the temporal area and attempting filler procedures in this area is recommended. In the cadaver study, the dyed gelatin was confirmed to be accurately placed in the superficial temporal fat pad (Figure 6). Within the deep temporal fascia containing the superficial temporal fat pad, structures such as the middle temporal vein running parallel above the fat pad and the sentinel vein connecting vertically to the middle temporal vein can be identified. Understanding the anatomical locations allows safe procedures in the superficial temporal fat pad without damaging these blood vessels (Figure 7). Beneath the superficial temporal fat pad containing the gelatin, the deep layer of the deep temporal fascia can be lifted to reveal the deep temporal fat pad located above the temporalis muscle, also known as the superior lobe or temporal extension of the buccal fat (Figure 8).

### 3.3. Subprocerus and Subnasalis Space

The skin and soft tissue layers of the nose, viewed from the top, are composed of five layers: the skin–subcutaneous fat layer, the fibromuscular layer connecting to the SMAS and including the nasalis muscle at the bridge of the nose and the procerus muscle at the root of the nose, the submuscular fat layer, and the nasal bone and cartilage layer. The proximal part of the nose, including the root, has dynamic properties, allowing smooth movement of the layers above and below the fibromuscular layer, enabling filler procedures in both the superficial and deep layers. However, the distal part, including the nasal tip, has a tight attachment of the fibromuscular layer to the skin due to the dense collagen fiber and fat cell composition of the subcutaneous fat layer, especially around the boundary of the lateral nasal cartilage and alar cartilage, known as the scroll area. Therefore, for safe volumization, it is recommended to inject fillers into the subprocerus space and the subnasalis space beneath the muscles (Figure 9 and Figure 10) [35,36]. During the procedure, the practitioner should lift the skin sufficiently to include the procerus muscle or nasalis muscle below the subcutaneous tissue layer and inject the filler deeply beneath the muscle.

### 3.4. Preseptal Space of Upper Eyelid

When there is a sunken upper eyelid, it is challenging to directly inject into the orbital septum to enhance the volume of the septal fat due to concerns about damaging the blood vessels within the septal fat and the levator muscle, which raises the eyelid. Therefore, it is advisable to enhance the volume of the deep fat layer above the septum, in the same plane as the ROOF (Retro-Orbicularis Oculi Fat), located below the eyebrow. However, in reality, the deep fat layer becomes very thin in sunken eyelids, making it difficult to distinguish the fat layer [37,38]. Therefore, the preseptal space between the orbicularis oculi muscle, corresponding to the midfacial SMAS layer, and the septal wall should be targeted. This space allows the safe volumization of the sunken area without concerns about bleeding from the supraorbital artery and nerve or the palpebral vascular arcade present in the eyelid area. When the skin and subcutaneous fat tissue of the cadaver are lifted to reveal the orbicularis oculi muscle, the gelatin injected into the preseptal space is not visible (Figure 11A). However, lifting the muscle reveals the dyed gelatin along the orbital rim margin and under the deep fat, including the ROOF located below the eyebrow (Figure 11B). Further dissection reveals the gelatin accurately placed in the preseptal space above the orbital septum.

### 3.5. Suborbicularis Space of Tear Trough

The medial side of the orbit where the tear trough is located does not contain a deep fat layer, such as the SOOF, beneath the orbicularis oculi muscle. Therefore, the subSMAS space lies between the orbicularis oculi muscle, corresponding to the SMAS, and the periosteum. In reality, the orbicularis oculi muscle adheres flatly to the orbital bone, creating insufficient space for injection, resulting in the gelatin being injected not only beneath the muscle but also within the muscle itself. Figure 12A shows only the muscle, with the gelatin injected beneath it faintly visible. However, lifting the muscle reveals the dyed gelatin clearly (Figure 12B) [37,39,40].

### 3.6. Prezygomatic Space

For patients with a flat appearance due to hollowing in the anterior cheek area, apple cheek procedures are recommended. In this case, while Western patients typically receive periosteal layer injections due to bone volume reduction with age, most Korean patients, except those with congenitally deficient zygomatic bones, experience volume loss in the soft tissues above the bone rather than bone volume reduction with age, resulting in a flat anterior cheek appearance. Therefore, procedures to enhance the volume of this soft tissue area are necessary. The midfacial region, unlike the forehead or lower face, has well-developed superficial and deep fat compartments above and below the SMAS layer. Hence, for deep-layer procedures, fillers can be injected directly into the deep fat layer when minimal volumization is needed or into the prezygomatic space between the deep fat layer and the periosteum when significant volumization is required. This space is safe for procedures, as it does not contain important structures such as blood vessels and nerves.

The prezygomatic space is usually formed above the zygomatic bone body. Its floor is the origin of the zygomatic muscles, the roof is the orbicularis oculi muscle line, the upper border is the orbicularis retaining ligament, and the lower border is the zygomatic ligaments. This space communicates with the lower temporal space of the temple area through the temporal tunnel and can be divided into layers, as shown in Table 2. In cadaver studies, dyed gelatin placed in the prezygomatic space on the bone, corresponding to the preperiosteal space in the malar region, can be observed. The fat in this space appears whiter and looser compared to the yellow SOOF above it (Figure 13) [37,41,42].

### 3.7. Premaxillary and Ristow’s Spaces

The premaxillary space is bordered by the zygomatico-cutaneous ligaments and the lateral part of the maxillary ligament or by the angular artery, an extension of the facial artery running along the side of the nose, and the angular vein, an extension of the facial vein running along the nasojugal groove, similar to the prezygomatic space, representing a space between the deep fat layer and the periosteum (Figure 14). Similarly, Ristow’s space, used for paranasal depression correction procedures, is a midfacial plane created between the medial part of the deep medial cheek fat and the periosteum in cases of sunken canine fossa. Cadaver images show dyed gelatin in Ristow’s space in the paranasal region of the nasolabial fold (Figure 15).

Since the premaxillary space and Ristow’s space are on the same plane, fillers injected beneath the nasolabial fold for correction may migrate across the fold to the upper part of the fold. Therefore, when performing filler procedures in the paranasal region, it is important to press the upper part of the nasolabial crease to prevent the filler from moving to the upper part of the fold.

### 3.8. Preparotid and Premasseteric Space

East Asians typically have more prominent zygomatic arches and cheekbones compared to Westerners, which can make the lateral cheek hollow below the zygomatic arch appear more pronounced. For individuals with a clear lateral cheek hollow, even without a thin face, this hollow is emphasized due to the strong retaining ligaments, such as the zygomatic ligaments and the parotid-masseteric ligament, along with the ligaments covering the platysma, pulling the skin downward (Figure 16). In such cases, injecting fillers into the subcutaneous fat layer may not effectively enhance the volume, and it may result in a bulging or hard appearance without lifting the area upward. This is because the subcutaneous layer is not the appropriate injection layer in areas with developed ligamentous structures. Instead, fillers should be injected into the deeper preparotid and premasseteric space. The preparotid and premasseteric space refers to the area above the upper half of the masseter muscle covered by the parotid gland and above the lower half of the masseter muscle. This space is similar to the upper and lower temporal spaces formed above the deep fascia of the temporalis muscle, as it lies above the deep fascia of the masseter muscle. The floor of this space is the parotid-masseteric fascia, the ceiling is the platysma, the posterior boundary is defined by the firm platysma-auricular ligament, and the anterior boundary is defined by the masseteric-cutaneous ligament (Table 3) [16,34,43].

This space is typically divided into superior, middle, and inferior compartments by fibrous septa, which can be easily dissected with blunt dissection. Each compartment is relatively safe, and the locations of important structures are at the boundaries of these compartments. In the cadaver study, the lower branches of the facial nerve’s zygomatic branch were found to pass shallowly above the superior compartment under the inner part of the upper masseteric ligament. The parotid duct passes between the superior and middle compartments, and the buccal branch of the facial nerve passes between the middle and inferior compartments. The marginal mandibular branch of the facial nerve runs below the inferior compartment along the jawline (Figure 17).

Fortunately, structures such as the parotid duct and the buccal branch of the facial nerve, emerging from the parotid gland, are covered by the deep fascia within the space, ensuring safety [44,45]. Therefore, unless a thick needle directly pierces deep structures, filler procedures in the space between the superficial and deep fascia can be performed safely. However, structures passing through the preparotid and premasseteric space typically emerge from shallow areas near the anterior border of the masseter muscle, which forms the boundary between the lateral and anterior face, through the masseteric ligaments. Hence, care is needed when performing procedures in the deep fat layer of the anterior face near the boundary between the lateral face and anterior face, where the branches of the facial nerve approach the SMAS layer [46,47].

When correcting severe grooves created by ligamentous structures during filler procedures, it is often necessary to create some space through partial tunneling of the tough ligamentous tissue. Similarly, for lateral cheek hollow procedures, partial tunneling through the ligaments beneath the zygomatic arch and between the masseteric-cutaneous ligament using a cannula creates space for filler injection. After creating space, injecting filler into the preparotid and premasseteric space between the SMAS and the deep fascia, as shown in Figure 18, can smooth out the sunken areas without highlighting the boundaries, providing sufficient volume enhancement (Figure 19).

### 3.9. Prebuccal Space

In the buccal region, filler can be injected into the buccal space containing buccal fat or the outer space, known as the prebuccal space. The buccal space is one of the deep fascial spaces formed below the deep fascia, including the submandibular gland. This space and its contents, the buccal fat, facilitate the movement of the nasolabial segment in the mid-cheek and cushion the excessive motion of the jaw. The area occupied by buccal fat is broader than typically imagined, extending from the upper part of the mandible to the temporal region. It is divided into superior, middle, and inferior lobes, each enclosed by a separate capsule, with the inferior lobe often referred to as the buccal fat pad (Figure 20) [48,49,50].

The buccal space, containing the buccal fat pad, is an anatomically safe area. The parotid duct runs above it (between the middle and inferior lobes), and the facial artery and the marginal mandibular branch of the facial nerve pass along its floor, always traversing below the inferior lobe near the mandible. In youth, the buccal space is located above the oral commissure level and medial to the anterior border of the masseter muscle. As one ages and the space enlarges, the buccal fat may prolapse below the oral commissure level, approaching the lower anterior border of the masseter muscle, thereby deepening the marionette lines and jowls. The buccal space, fundamentally positioned below the medial and lateral parts of the deep medial cheek fat in the midface, has the boundaries outlined in Table 4 [48,49].

For buccal cheek hollow correction, directly enhancing the volume of the buccal fat pad encapsulated in a thin capsule can cause the inner cheek to bulge more than the outer cheek, which is undesirable. Therefore, it is more effective to enhance the volume in the prebuccal space, which is the space between the capsule enclosing the buccal fat pad and the SMAS, with a smaller amount of filler. The facial nerve’s buccal branch typically runs within the capsule, so there is no need to worry about damaging it unless a thick needle directly pierces this area. In cadaver studies, dyed gelatin placed in the prebuccal space between the SMAS and the buccal fat pad capsule can be observed (Figure 21).

### 3.10. Subdepressor Anguli Oris Space

Among the muscles forming the modiolus, the depressor anguli oris muscle and the superficial fibers of the orbicularis oris muscle are the most superficial. When there is a significant height difference between these superficial fibers and the deeper fibers of the orbicularis oris muscle, a commissural line appears diagonally from the mouth corner, marking the boundary between the cheek and the lip area. As one ages, the lateral lower lip fat compartment among the three superficial fat compartments beneath the lip (two lateral and one central) may become sunken, intensifying this commissural line and causing the mouth corner to droop. This pronounced boundary, formed by the differences in tissue thickness, is referred to as the melolabial fold [51].

To mitigate the difference and smooth out the boundary, it is effective to fill the space below the depressor anguli oris muscle, including the deep fat. The sunken area beneath the melolabial fold can be enhanced to soften the boundary (Figure 22).

### 3.11. Premental Space

When enhancing the volume of the chin using fillers, injecting directly onto the chin bone after penetrating the soft tissues beneath the skin may require a large amount of filler. To achieve sufficient volume with less filler, it is better to inject into the premental space, which includes the deep fat layer between the mentalis muscle attached to the chin skin and the bone. In cadaver studies, dyed gelatin placed in the premental space between the mentalis muscle and the mentum can be observed (Figure 23).

This concludes the detailed exploration of each subSMAS adipofascial space in the face for safe and effective filler procedures. By accurately identifying and utilizing these spaces, practitioners can perform procedures with greater precision, enhancing facial volume while minimizing the risk of damaging important structures.

## 4. Discussion

Even though deep spaces are the recommended areas for filler injections, there remains a risk of vascular invasion and penetration of anatomical structures when approaching these regions. This can lead to several disadvantages and side effects, including vascular complications, which can cause tissue necrosis or embolism if a blood vessel is blocked. Bruising and swelling are common, as the injection process can damage blood vessels, and these effects may persist for several days. Infections are a risk due to the breach of skin, potentially resulting in abscess formation or cellulitis. Nerve damage may occur from improper injection techniques, causing pain, numbness, or altered sensation. Over time, the body might react to the filler material by forming granulomas, small painful lumps that require further treatment. Incorrect filler placement can lead to asymmetry or irregularities, necessitating additional procedures to correct the appearance. Although rare, allergic reactions to filler materials can cause redness, swelling, and itching. These potential complications highlight the importance of having filler injections performed by a skilled and experienced practitioner to minimize risks (Table 5) [52,53,54,55,56,57,58,59].

As described in a recent study by Lim et al. [60], overfilled syndrome is attributed to the excessive accumulation of filler in one region rather than its distribution in subSMAS spaces and superficial layers. Preventing and managing facial overfilled syndrome requires a deep understanding of facial anatomy, including the role of deep fat compartments in maintaining facial volume and contour, the interaction between facial muscles and the underlying bone structures for natural filler support, and pressure dynamics, where techniques like subcision and botulinum neurotoxin help manage high-pressure areas to avoid unnatural results and complications, ensuring natural and harmonious facial aesthetics.

Also, according to the recent anatomical study by Cotofana et al. [61], the transverse facial septum, as described in the article, is a significant anatomical structure originating from the underside of the zygomaticus major muscle and attaching to the maxilla. It acts as a transversely running boundary between the buccal space and the deep midfacial fat compartments. This septum is involved in facial dynamics, where the contraction of the zygomaticus major muscle during expressions such as smiling can increase tension on the septum, leading to a cranial shift in the deep midfacial fat compartments. This shift influences the anterior projection of the midface, which is crucial for understanding and managing facial aesthetics and conditions like facial overfilled syndrome.

The recent study of Yi et al. [62] emphasizes the importance of subSMAS injection for hyaluronic acid fillers in treating subzygomatic arch depression. Injecting fillers into the subSMAS layer provides enhanced volume addition by targeting the atrophied deep fat layers, thus addressing severe depression more effectively. This technique reduces surface irregularities and avoids the superficial bumpy shaping and undulations associated with single-plane injections above the SMAS. The dual-plane injection method ensures controlled filler spread by injecting both below and above the SMAS layer, preventing unwanted migration and ensuring a smoother contour. Deep-plane injections also minimize the risk of filler particles shifting toward the premasseteric region, maintaining the desired aesthetic outcome. Additionally, this approach decreases the likelihood of irregularities and complications, such as inferior migration and projection loss, commonly seen with high-volume injections in the superficial plane.

According to the recent study by Hong et al. [63], the advantage of the subSMAS injection plane, as argued by the authors, lies in its ability to achieve both volume restoration and natural lifting effects while minimizing complications and avoiding unnatural appearances. Injecting firm fillers into the subSMAS layer increases the density of the subSMAS tissue, leading to a tightening effect on the SMAS layer. This tension not only supports the SMAS layer but also creates a traction effect that extends to adjacent areas, resulting in a natural lifting effect. This strategic placement capitalizes on the anatomical properties of the SMAS, allowing for effective facial contouring and volumization without the risk of filler migration and irregularities that can occur with superficial injections. By understanding and utilizing the distinct mechanical properties and anatomical variability of facial tissues, the subSMAS injection technique ensures more precise and harmonious aesthetic outcomes.

The identification and utilization of subSMAS spaces for filler procedures represent a significant advancement in aesthetic medicine, particularly for enhancing safety and efficacy. This study emphasizes the importance of understanding the intricate anatomy of facial spaces to perform filler injections more effectively [1,15,64,65].

The anatomical distinction of these spaces, primarily those between the deep fascia and the superficial fascia, provides a framework for more precise and independent muscle movements. This distinction is crucial, as it allows specific muscles to operate without interfering with the movement of adjacent muscles, thereby preserving natural facial expressions. For instance, the independent movement of the orbicularis oculi and orbicularis oris muscles while the zygomaticus major and minor muscles function illustrates this concept. This independence is particularly valuable in aesthetic procedures where maintaining natural expressions is essential.

Moreover, the relative safety of these spaces is underscored by the anatomical passage of blood vessels and nerves predominantly through the boundaries rather than the interiors. This anatomical feature minimizes the risk of vascular or neural damage during filler injections, enhancing the safety profile of these procedures [66,67].

This study also highlights the unique considerations for East Asian patients, whose thicker and more robust skin structures necessitate a deeper injection approach. The increased density and toughness of the SMAS and retinacular cutis in East Asians make superficial injections less effective, thus requiring a more profound understanding of the subSMAS spaces for optimal volumization.

The cadaver study using dyed gelatin provided empirical evidence of the existence and characteristics of these subSMAS spaces. The successful injection and subsequent dissection confirmed that the dyed gelatin accurately filled the intended spaces without compromising vital structures. This validation supports the practical application of these anatomical insights in clinical settings.

Each identified space, from the subgalea-frontalis space to the premental space, serves a specific purpose in filler procedures. For example, the subgalea-frontalis space allows for safe injections in the forehead area without damaging the supraorbital and supratrochlear arteries and nerves. Similarly, the interfascial and temporalis space in the temple region facilitates volumization without the risk of creating uneven surfaces.

Schelke et al. discuss the risks associated with filler injections in the preauricular space, particularly due to the close proximity to the parotid gland. This region, though increasingly used for augmenting facial width and enhancing the mandibular angle, presents unique challenges. Injecting fillers in this area can inadvertently lead to the filler material being deposited inside the parotid gland, often unnoticed by both the injector and the patient. This can result in inflammation, edema, pain, tender nodules, and even abscess formation. Studies have shown that such injections can cause subclinical signs of inflammation in the parotid gland, detectable through sonography even when no visible symptoms are present. The limited space for injection, often just 2–4 mm thick, further complicates the procedure, increasing the likelihood of complications. It is recommended to use ultrasound imaging prior to treatment to measure the subcutaneous depth and guide the injection process accurately to avoid such adverse effects.

This study sheds light on the challenges and solutions for filling areas with significant hollowing, such as the temporal region and the lateral cheek hollow. For instance, injecting into the superficial temporal fat pad rather than beneath the temporalis muscle avoids complications related to muscle contractions and filler migration.

Furthermore, the study elucidates the practical aspects of injecting fillers into spaces like the prezygomatic space, premaxillary space, and Ristow’s space. These areas, devoid of critical structures, offer safe zones for substantial volumization, addressing concerns of midface volume loss effectively.

This study, while comprehensive in its approach to identifying and delineating subSMAS (sub-superficial musculoaponeurotic system) spaces, has several limitations that must be acknowledged. Firstly, the study was conducted using cadavers from East Asia, specifically Korean individuals, which may limit the generalizability of the findings to other populations with different anatomical characteristics. The sample size of 10 cadavers, while sufficient for preliminary findings, may not provide a comprehensive representation of the variability in facial anatomy across a broader population.

Additionally, the cadavers used in the study were not diversified in terms of age and sex. This lack of diversity could affect the applicability of the results to different age groups and genders, as facial anatomy can vary significantly with age and between sexes. Furthermore, the study was conducted on cadavers, which may not fully replicate the conditions and responses of living tissues during filler procedures. The dynamic nature of living tissues, including blood flow and muscle movement, is not accounted for in cadaver studies.

The use of dyed gelatin injections, while effective for visualizing spaces in cadaver studies, may not perfectly mimic the behavior and distribution of actual dermal fillers used in clinical practice. The study also does not address the long-term outcomes of filler injections in the identified subSMAS spaces. Future research is needed to evaluate how fillers behave over time in these anatomical zones and their long-term safety and efficacy.

Moreover, the study’s focus on East Asian anatomy may not account for anatomical differences present in other ethnic groups. Variations in skin thickness, tissue density, and facial structure across different ethnicities need further exploration. By recognizing these limitations, we highlight the need for further research to validate and expand upon our findings, incorporating a more diverse range of subjects and considering the dynamic properties of living tissues. This will help to ensure that the anatomical insights provided are broadly applicable and effective in enhancing the safety and efficacy of facial filler procedures across different populations.

## 5. Conclusions

This study successfully identifies and delineates the subSMAS (sub-superficial musculoaponeurotic system) spaces crucial for safe and effective facial filler procedures. By using dyed gelatin injections in cadaver studies, we have provided clear anatomical evidence of these spaces, confirming their existence and safety for clinical applications.

The findings highlight the importance of understanding the intricate anatomy of facial spaces to enhance the precision and safety of filler injections. This knowledge is particularly valuable for East Asian patients, whose thicker and more robust skin structures necessitate deeper injection techniques.

The detailed exploration of key spaces such as the subgalea-frontalis, interfascial and temporalis, and prezygomatic spaces offers a comprehensive guide for clinicians. These spaces, devoid of critical structures like major blood vessels and nerves, represent safe zones for volumization and aesthetic enhancement.

Furthermore, this study emphasizes the need for precise anatomical knowledge to perform filler procedures effectively, minimizing the risk of complications and maintaining natural facial expressions. The subSMAS spaces facilitate independent muscle movement, which is essential for preserving the dynamic nature of facial expressions.

In summary, this anatomical study provides a foundational guide for clinicians to perform precise and safe filler injections. By leveraging the detailed map of subSMAS spaces, practitioners can improve facial aesthetics while minimizing risks. Future research should focus on the dynamic interactions of these spaces during facial movements and the long-term outcomes of filler procedures in these anatomical zones.

## Figures and Tables

**Figure 1 diagnostics-14-01452-f001:**
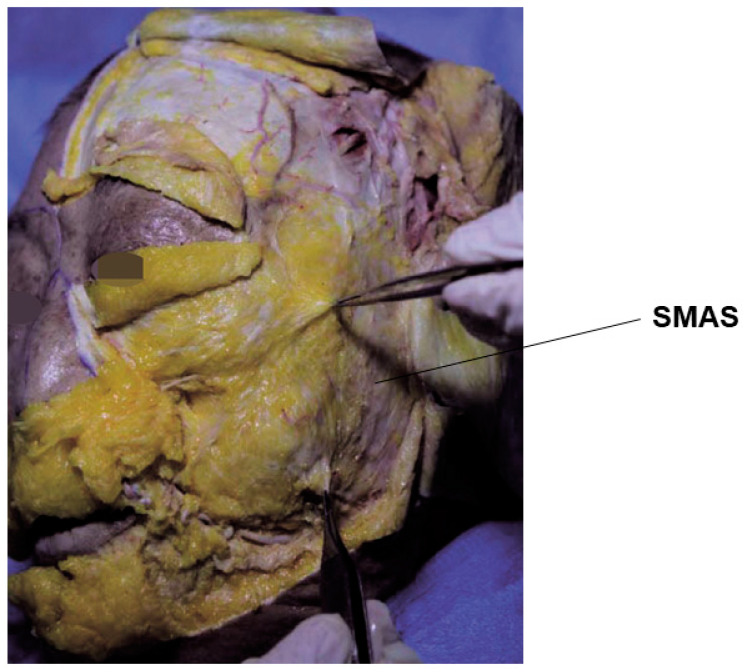
This figure represents the thick SMAS in the lateral cheek region.

**Figure 2 diagnostics-14-01452-f002:**
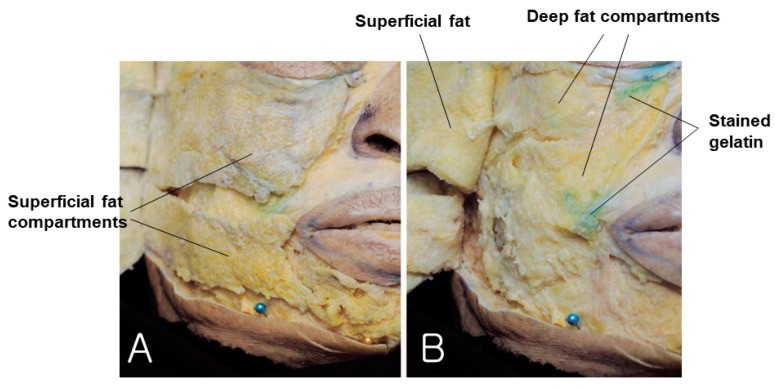
This figure represents stained gelatin in subSMAS spaces beneath superficial fat compartments. The panel (**A**) shows superficial fat compartments and panel (**B**) shows deep fat compartment.

**Figure 3 diagnostics-14-01452-f003:**
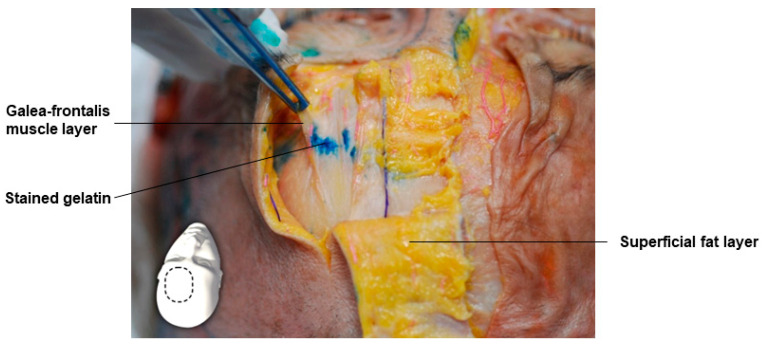
This figure represents gelatin in the subgalea-frontalis space.

**Figure 4 diagnostics-14-01452-f004:**
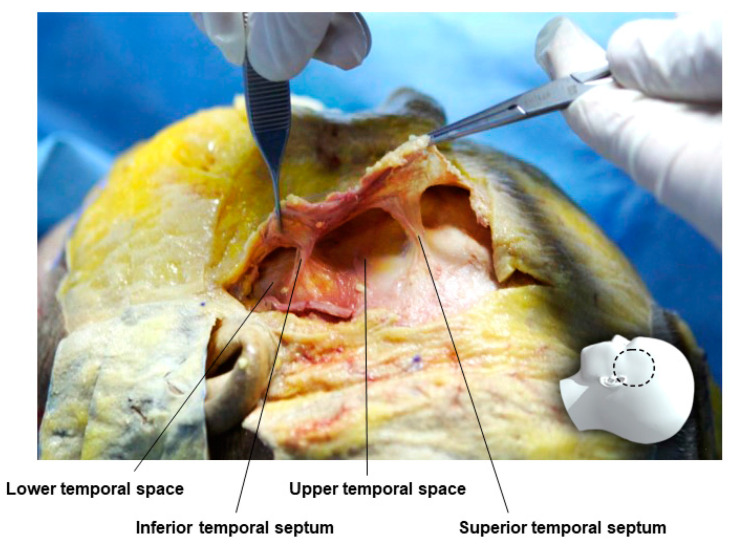
This figure represents the upper and lower temporal spaces.

**Figure 5 diagnostics-14-01452-f005:**
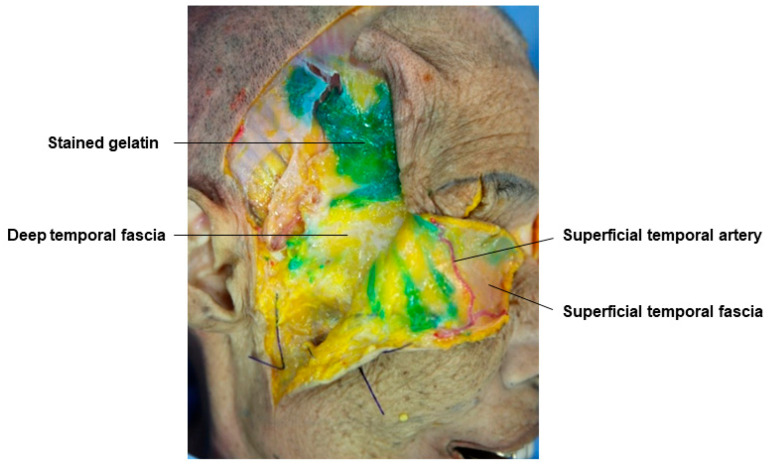
This figure represents gelatin in the space between the superficial and deep temporal fascia.

**Figure 6 diagnostics-14-01452-f006:**
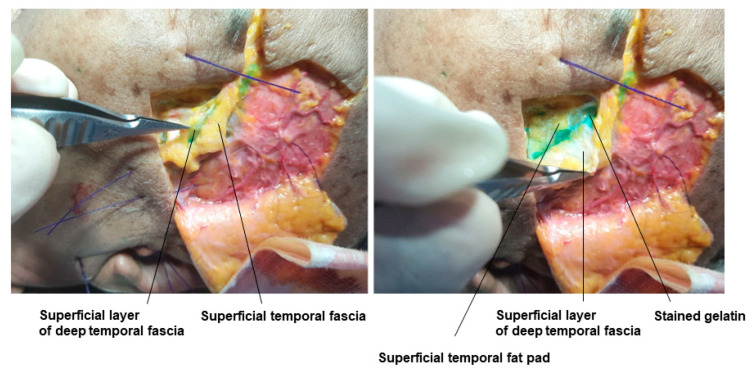
This figure represents gelatin in the superficial temporal fat pad.

**Figure 7 diagnostics-14-01452-f007:**
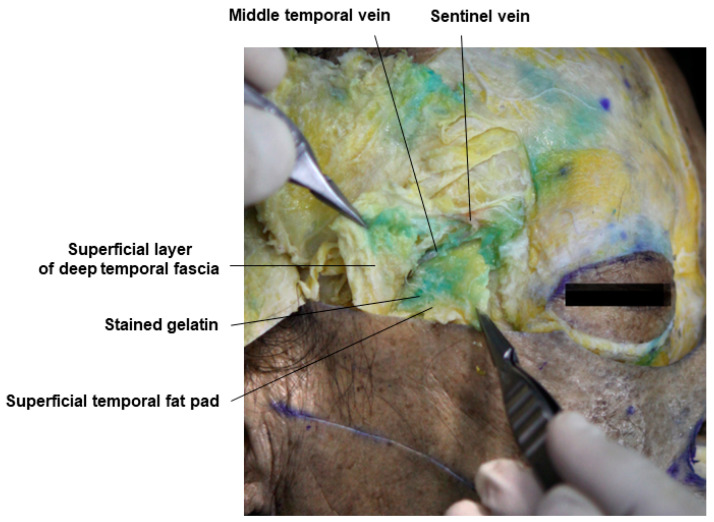
This figure represents the middle temporal vein and sentinel vein.

**Figure 8 diagnostics-14-01452-f008:**
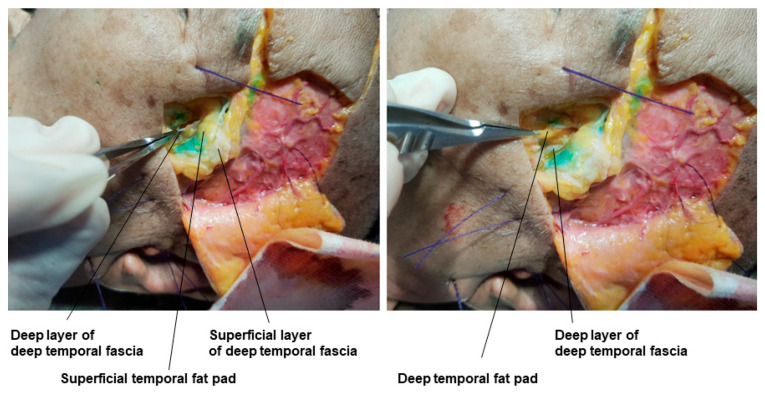
This figure represents the deep temporal fat pad under the deep temporal fascia.

**Figure 9 diagnostics-14-01452-f009:**
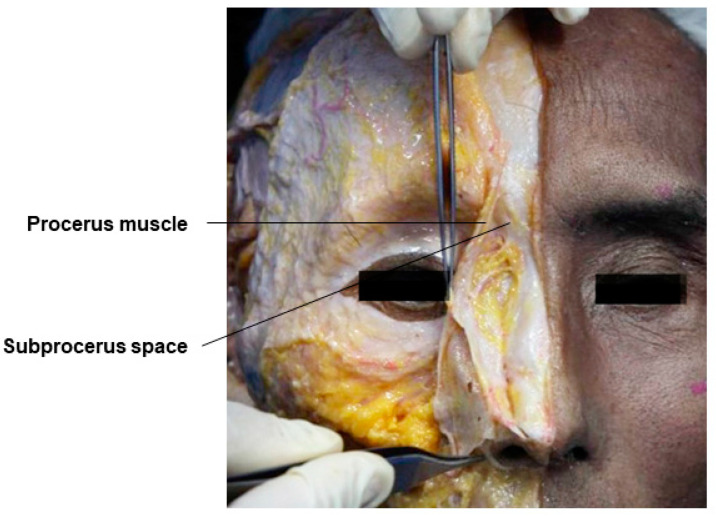
This figure represents the subprocerus space of the nose.

**Figure 10 diagnostics-14-01452-f010:**
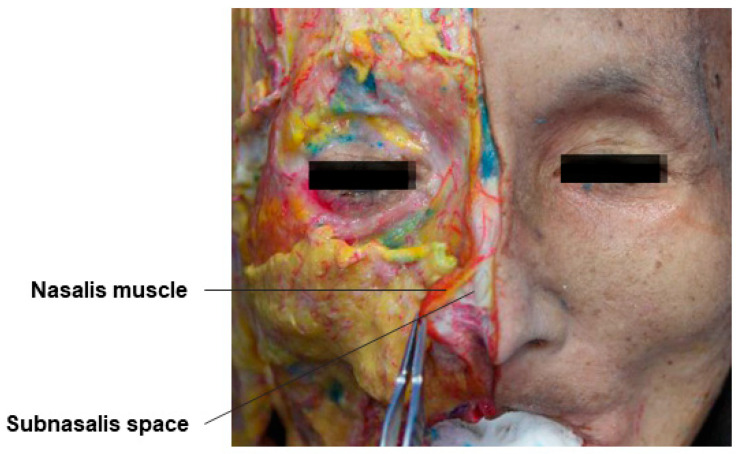
This figure represents the subnasalis space of the nose.

**Figure 11 diagnostics-14-01452-f011:**
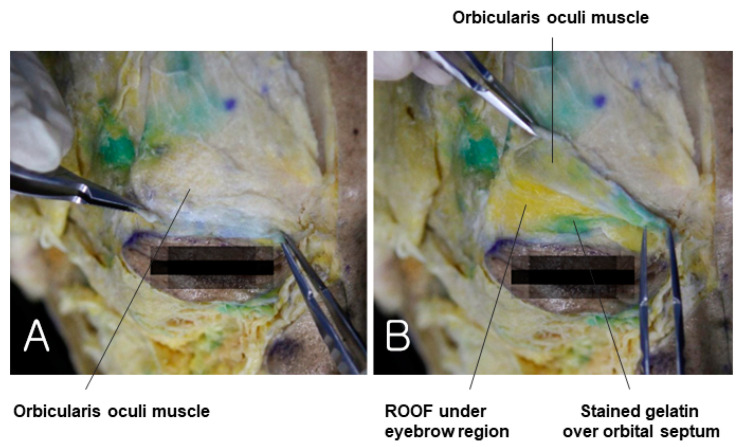
This figure represents the ROOF (Retro-Orbicularis Oculi Fat) beneath the orbicularis oculi muscle. The panel (**A**) shows orbicularis oculi muscle and panel (**B**) demonstrates removed orbicularis oculi muscle.

**Figure 12 diagnostics-14-01452-f012:**
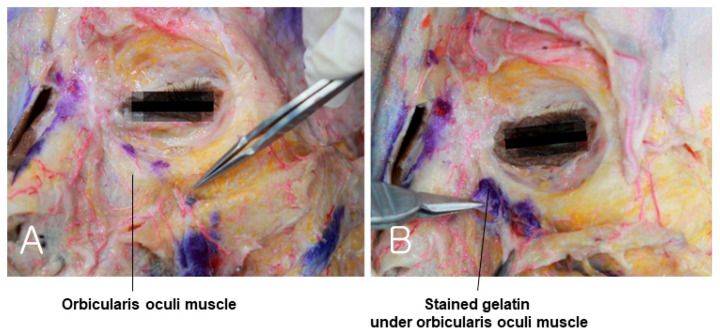
This figure represents gelatin under the orbicularis oculi muscle in the tear trough region. The panel (**A**) shows orbicularis oculi muscle and panel (**B**) demonstrates gelatin under orbicularis oculi muscle.

**Figure 13 diagnostics-14-01452-f013:**
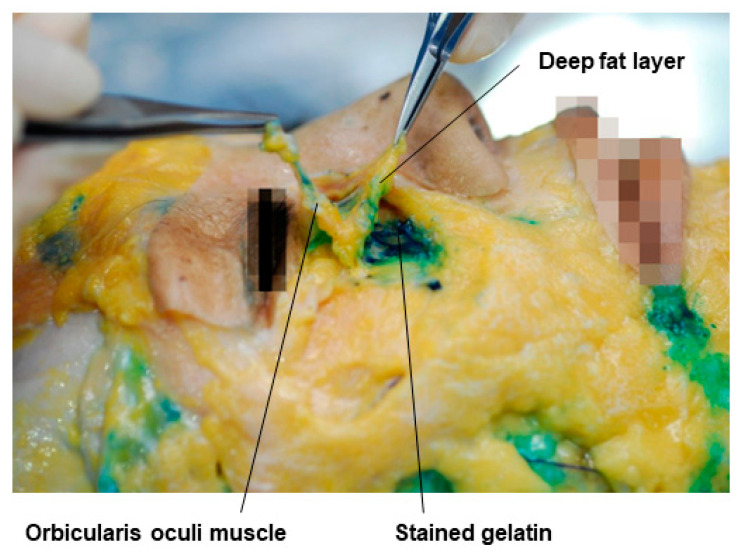
This figure represents gelatin in the prezygomatic space.

**Figure 14 diagnostics-14-01452-f014:**
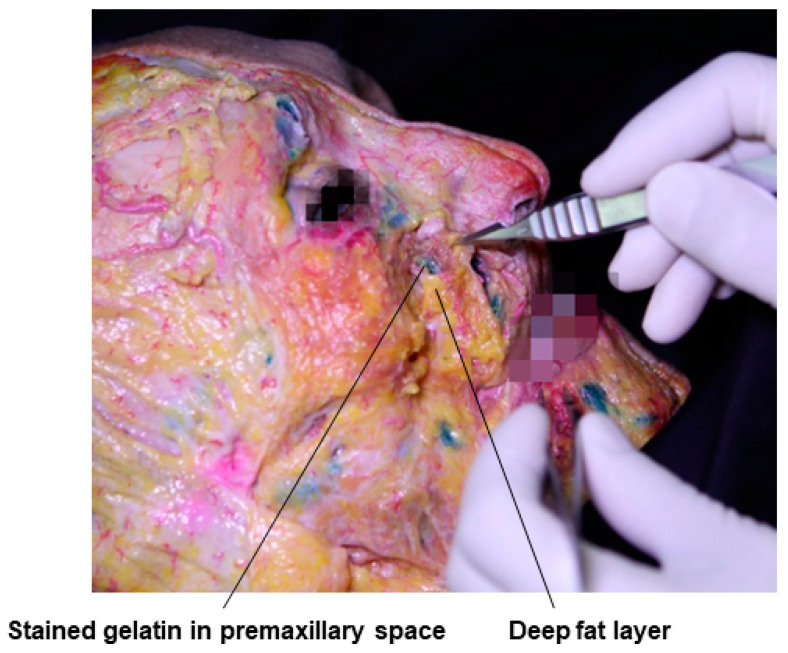
This figure represents gelatin in the premaxillary space.

**Figure 15 diagnostics-14-01452-f015:**
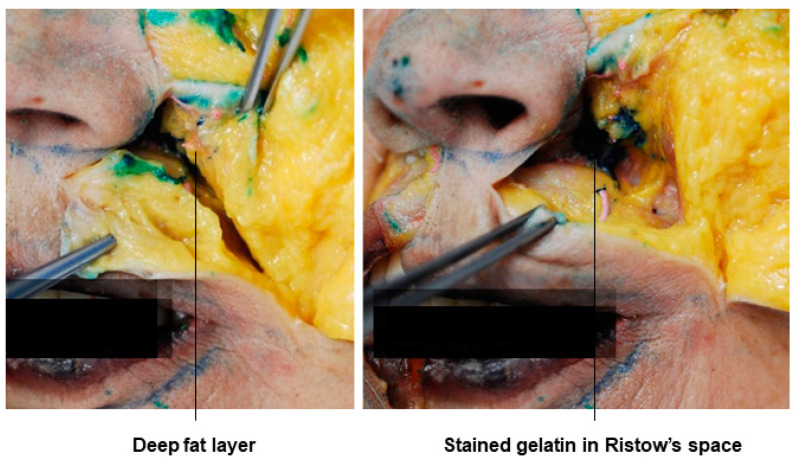
This figure represents gelatin in Ristow’s space in the paranasal region.

**Figure 16 diagnostics-14-01452-f016:**
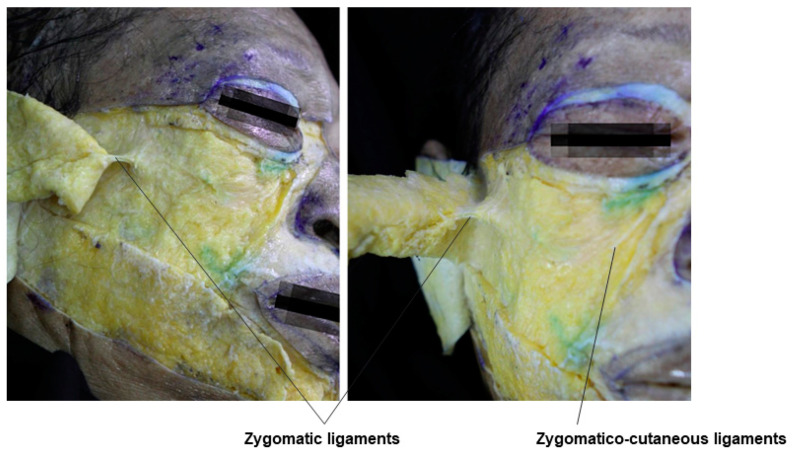
This figure represents the zygomatic ligaments in the lateral cheek region.

**Figure 17 diagnostics-14-01452-f017:**
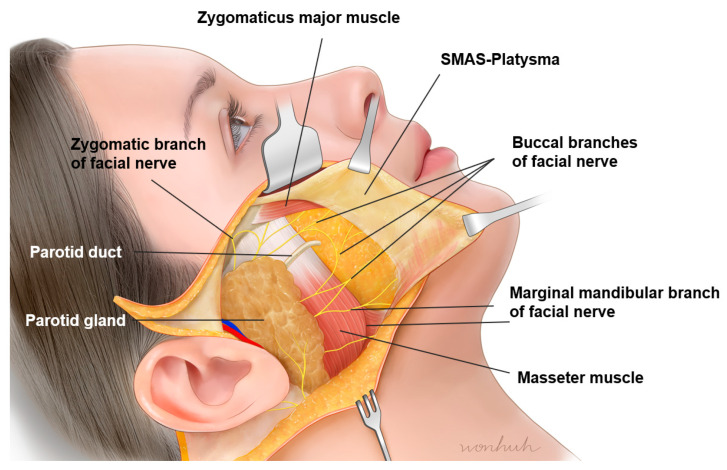
This figure represents the anatomical structures between the compartments of the preparotid and premasseteric space.

**Figure 18 diagnostics-14-01452-f018:**
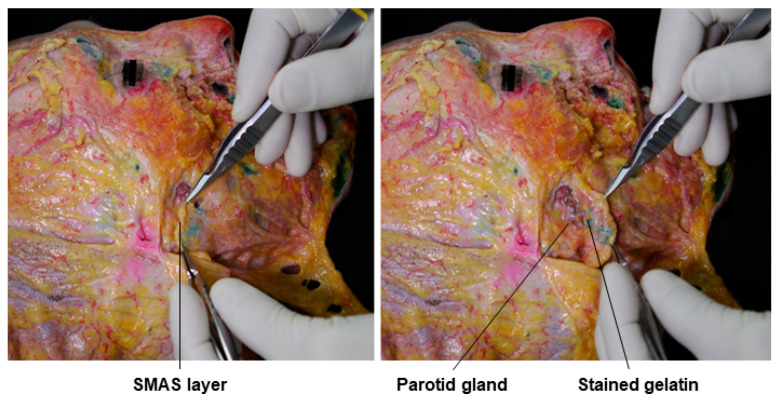
This figure represents gelatin in the preparotid and premasseteric space.

**Figure 19 diagnostics-14-01452-f019:**
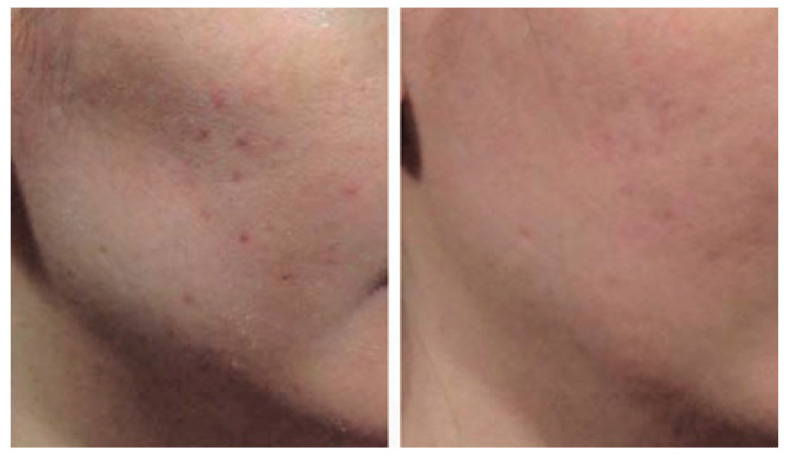
This figure represents the before and after treatment of lateral cheek hollowness.

**Figure 20 diagnostics-14-01452-f020:**
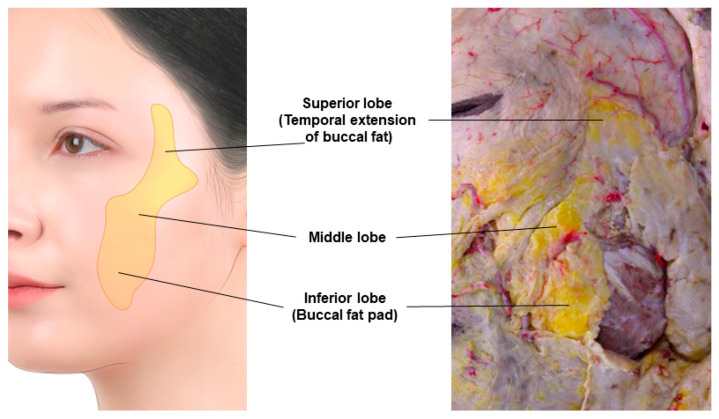
This figure represents the extensions of buccal fat.

**Figure 21 diagnostics-14-01452-f021:**
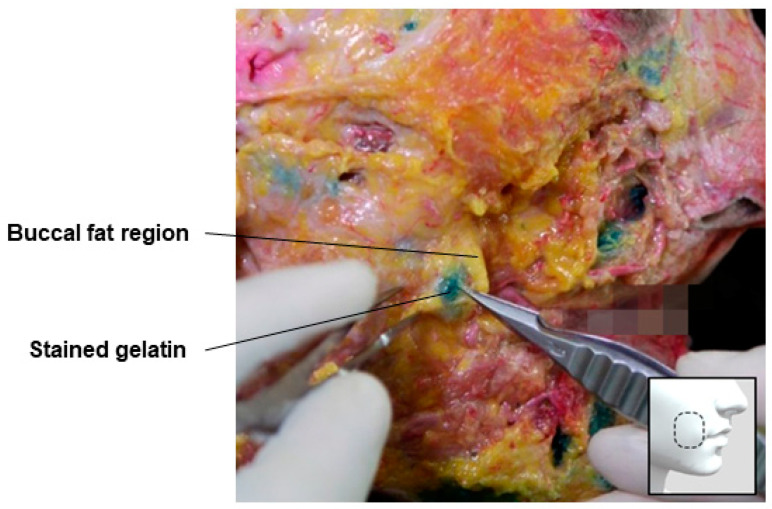
This figure represents gelatin in the prebuccal space.

**Figure 22 diagnostics-14-01452-f022:**
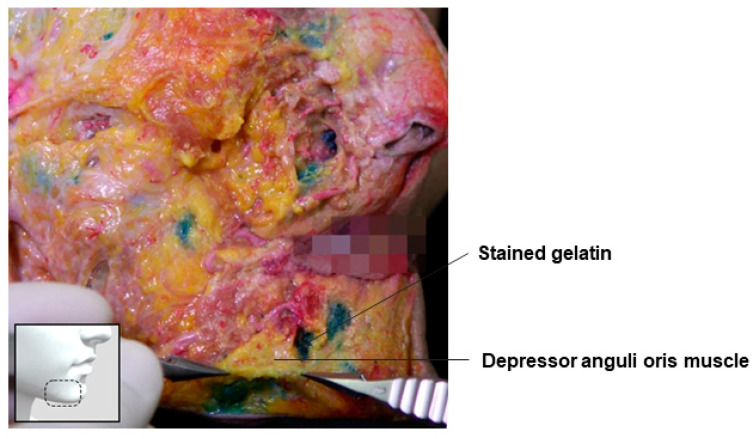
This figure represents gelatin in the subdepressor anguli oris space.

**Figure 23 diagnostics-14-01452-f023:**
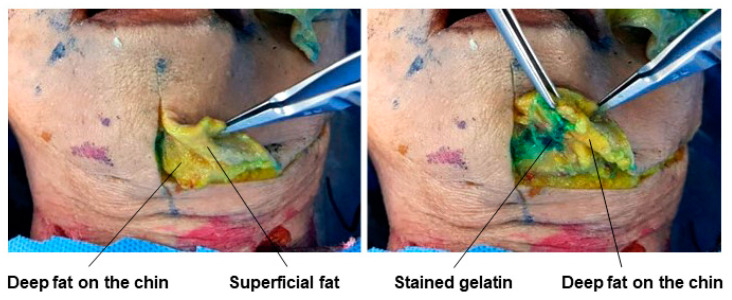
This figure represents gelatin in the premental space.

**Table 1 diagnostics-14-01452-t001:** Boundaries and indications of the possible subSMAS adipofascial spaces.

Boundary	Description	Indication
Subgalea-frontalis space	Area under the frontalis muscle	Flat forehead
Interfascial & pretemporalis space	Compartment between STF and DTF, STFP between the superficial and deep layer of DTF	Temple augmentation
Subprocerus space	Area under the procerus muscle	Glabellar depression
Preseptal space	Area that overlies the orbital septum below the orbital retaining ligament in the upper eyelid	Sunken eyelid
Suborbicularis space	Area under the orbicularis oculi muscle (ROOF of the eyebrow, medial part of the SOOF and lateral portion of deep medial cheek fat)	Infraorbital groove and hollowness
Subnasalis space	Area under the nasalis muscle	Flat nose
Prezygomatic space	Area that overlies the body of the zygoma; Its floor covers the origin of the zygomaticus muscle	Apple cheek
Premaxillary space	Area that overlies the maxilla bone - Its floor covers the origin of the levator labii superioris muscle	Anteromedial cheek hollowness
Ristow’s space (pyriform space)	Area that overlies the canine fossa under the medial part of the DMCF in the paranasal region	Paranasal depression and nasolabial fold
Preparotid & premasseteric space	Area that overlies the parotid gland and the lower half of the masseter	Lateral cheek hollowness
Prebuccal space	Area that overlies the capsule of the buccal fat pad medial to the anterior border of the masseter	Buccal cheek hollowness
Subdepressor anguli oris space	Area including fat deposits under the depressor anguli oris muscle	Marionette line
Premental space	Area including fat deposits under the skin insertion of the mentalis muscle	Chin augmentation

**Table 2 diagnostics-14-01452-t002:** Seven layers of prezygomatic space.

Layer Number	Layer Description
1	Skin
2	Subcutaneous fat layer
3	Orbicularis oculi muscle
4	SOOF (Sub-Orbicularis Oculi Fat)
5	Deep fascia origin of zygomatic muscles
6	Preperiosteal (prezygomatic) fat layer
7	Periosteum

**Table 3 diagnostics-14-01452-t003:** Boundaries of the preparotid and premasseteric space.

Boundary	Description
The floor	The parotid gland and the lower half of the masseter muscle
The roof	The SMAS and platysma muscle
The posterior border	The anterior edge of the strong platysma-auricular ligament
The anterior border	The masseteric ligaments near the anterior edge of the masseter muscle

**Table 4 diagnostics-14-01452-t004:** Boundaries of the buccal space.

Boundary	Description
The floor	The buccinator muscle
The roof	Mimetic muscles and the SMAS
The superior boundary	The maxillary ligament
The inferior boundary	Loose adhesion of the platysma muscle
The anterior boundary	The modiolus
The posterior boundary	The facial vein and the masseteric ligaments

**Table 5 diagnostics-14-01452-t005:** Table summarizing the disadvantages and side effects of filler injections.

Disadvantage/Side Effect	Description
Vascular complications	Unintentional injection into blood vessels can cause blockages, leading to tissue necrosis or embolism, which can be serious and sometimes life-threatening.
Bruising and swelling	The injection process can damage blood vessels, resulting in bruising and swelling, which may persist for several days.
Infection	Any breach of the skin carries a risk of infection, which can lead to complications such as abscess formation or cellulitis.
Nerve damage	Improper injection technique may damage nerves, causing pain, numbness, or altered sensation in the affected area.
Granulomas	Over time, the body may react to the filler material by forming granulomas, which are small lumps that can be painful and require further treatment.
Asymmetry and irregularities	Incorrect filler placement can result in asymmetry or irregularities, necessitating additional procedures to correct the appearance.
Allergic reactions	Although rare, allergic reactions to filler materials can occur, leading to redness, swelling, and itching.

## Data Availability

Data are available on request to corresponding author.

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
