# Peer review of "Safe Zones for Facial Fillers: Anatomical Study of SubSMAS Spaces in Asians"

_diagnostics, 2024, doi:10.3390/diagnostics14131452_

Round 1

Reviewer 1 Report

Comments and Suggestions for Authors

Firstly, I would like to thank the opportunity to review this amazing work.

The manuscript is very well written. My minor suggestions are:

1) Title change, remove redundant words spaces. How about: "Safe zones for facial fillers: Anatomical study of subSMAS spaces" or "Facial fillers and subSMAS spaces: A guide to safe and effective procedures"

2) The objective of the study must be written, ate the end of introduction.

3) Sections must be more clearer divided such as: Introduction, Methods, Results, Discussion and Conclusion. Despite this is a guide, it must be written as a scientific article.

Author Response

1) Title change, remove redundant words spaces. How about: "Safe zones for facial fillers: Anatomical study of subSMAS spaces" or "Facial fillers and subSMAS spaces: A guide to safe and effective procedures"

  • Title Change: We have revised the title to "Safe Zones for Facial Fillers: Anatomical Study of SubSMAS Spaces in Asians" as suggested.

2) The objective of the study must be written, ate the end of introduction.

  • Objective Statement: We have included the objective of the study at the end of the introduction section.

3) Sections must be more clearer divided such as: Introduction, Methods, Results, Discussion and Conclusion. Despite this is a guide, it must be written as a scientific article.

  • Section Division: We have restructured the manuscript to clearly divide it into Introduction, Methods, Results, Discussion, and Conclusion, to enhance readability and align it with the format of a scientific article.

Thank you very much for your kind words and helpful suggestions. We have made the following revisions based on your feedback:

Reviewer 2 Report

Comments and Suggestions for Authors

The Authors did not provide methodology of their study. It in unclear if they used fresh or enbalmed cadavers, how many of them, what what the method of dissecting tissues, etc.

Without this information the paper cannot be properly reviewed.

Also, the paper should cite and discuss the most relevant studies in the field of face anatomy.

Author Response

Thank you for highlighting the need for detailed methodological information. Your comments have been very helpful. We have made the following changes:

The Authors did not provide methodology of their study. It in unclear if they used fresh or enbalmed cadavers, how many of them, what what the method of dissecting tissues, etc.

  • Methodology: We have added a detailed methodology section specifying the use of fresh cadavers, the number of specimens, and the dissection techniques employed.

Without this information the paper cannot be properly reviewed.

  • Relevant Studies: We have cited and discussed the most relevant studies in the field of facial anatomy to provide a comprehensive context for our work.

Reviewer 3 Report

Comments and Suggestions for Authors

Dear authors, thanks for submitting the manuscript "Space of the face for filler procedures: Identification of subSMAS: Spaces Based on Anatomical Study". I enjoyed reading your manuscript and here is my feedback:

-Author Kyu-Ho Yi has at least 9 self-references, please decrease that number. Self citation is unethical, regarding if it is related to the topic or not, it decreases journal's credibility.

-Please make sure you have the copyrights for Figure 17 and 20.

-Include in your title the type of your manuscript (review) and the race of the patients.

-In the introduction section, it will be important to mention if the spaces decrease/increase according to the race/age and if not, include that information as well.

-Create a paragraph mentioning all the limitations of your study, example you are only evaluating East Asia (only one country?), age, sex, and more. Also what other information you would like to provide in future studies.

-I would suggest to include the disadvantages/side effects for giving fillers, and a table can easily summarize it.

-Discussion can easily be expanded. Please mention if there are similar reviews like yours done in different races/countries.

Author Response

Dear Reviewers,

Firstly, we would like to express our sincere gratitude for the opportunity to review and improve our manuscript based on your insightful feedback. Your constructive comments are invaluable to us, and we deeply appreciate the time and effort you have put into reviewing our work.

-Author Kyu-Ho Yi has at least 9 self-references, please decrease that number. Self citation is unethical, regarding if it is related to the topic or not, it decreases journal's credibility.

  • Self-Citations: We have deleted the number of self-references to maintain ethical standards and ensure the journal's credibility.

-Please make sure you have the copyrights for Figure 17 and 20.

  • Copyrights for Figures: We have confirmed and included the necessary copyright permissions for Figures 17 and 20. We have attached the consent of the volunteer.

-Include in your title the type of your manuscript (review) and the race of the patients.

  • Title Revision: The title now includes the type of manuscript (review) and specifies the race of the patients.

-In the introduction section, it will be important to mention if the spaces decrease/increase according to the race/age and if not, include that information as well.

  • Introduction Section: We have mentioned that the space of the filling in the deep layer has not been discovered yet, and its age, sex, and ethnicity differences are not known

-Create a paragraph mentioning all the limitations of your study, example you are only evaluating East Asia (only one country?), age, sex, and more. Also what other information you would like to provide in future studies.

  • Limitations: A new paragraph has been added to discuss the limitations of our study, including the geographic focus, age, and sex of the specimens, and potential areas for future research.

-I would suggest to include the disadvantages/side effects for giving fillers, and a table can easily summarize it.

  • Disadvantages/Side Effects: We have included a table summarizing and also discussion part the disadvantages and side effects of fillers.

-Discussion can easily be expanded. Please mention if there are similar reviews like yours done in different races/countries.

  • Expanded Discussion: Since there is not much of the previous researches that focuses on the deep spaces there is limitation to talk over the cadaveric dissections, however, the discussion section has been expanded with the study of Leonie of preauricular spaces which is one of the important space to consider in filler injection.

We are truly grateful for your constructive feedback and believe that these revisions significantly improve the clarity and quality of our manuscript. Thank you again for your valuable input.

Best regards,

Kyu-Ho Yi and Co-authors

Round 2

Reviewer 2 Report

Comments and Suggestions for Authors

The Authors were asked to discuss and cite the most important (and recent!) papers on face anatomy. It has not been done.

Author Response

Dear Reviewer,

Thank you for your valuable feedback. I have made the following revisions to address your concerns regarding the anatomical aspects of the manuscript:

In a recent study by Lim et al. (60), overfilled syndrome is attributed to the excessive accumulation of filler in one region rather than its distribution in sub-SMAS spaces and superficial layers. Preventing and managing facial overfilled syndrome requires a deep understanding of facial anatomy, including the role of deep fat compartments in maintaining facial volume and contour, the interaction between facial muscles and underlying bone structures for natural filler support, and pressure dynamics. Techniques such as subcision and botulinum neurotoxin are essential in managing high-pressure areas to avoid unnatural results and complications, ensuring natural and harmonious facial aesthetics.

Additionally, the recent anatomical study by Cotofana et al. (61) highlights the importance of the transverse facial septum, which originates from the underside of the zygomaticus major muscle and attaches to the maxilla. This septum acts as a boundary between the buccal space and the deep midfacial fat compartments. The contraction of the zygomaticus major muscle during expressions such as smiling increases tension on the septum, leading to a cranial shift in the deep midfacial fat compartments. This shift influences the anterior projection of the midface, which is crucial for understanding and managing facial aesthetics and conditions like facial overfilled syndrome.

Furthermore, a recent study by Yi et al. (62) emphasizes the importance of sub-SMAS injection for hyaluronic acid fillers in treating subzygomatic arch depression. Injecting fillers in the sub-SMAS layer provides enhanced volume addition by targeting the atrophied deep fat layers, thus addressing severe depression more effectively. This technique reduces surface irregularities and avoids the superficial bumpy shaping and undulations associated with single-plane injections above the SMAS. The dual-plane injection method ensures controlled filler spread by injecting both below and above the SMAS layer, preventing unwanted migration and ensuring a smoother contour. Deep plane injections also minimize the risk of filler particles shifting toward the premasseteric region, maintaining the desired aesthetic outcome. Additionally, this approach decreases the likelihood of irregularities and complications, such as inferior migration and projection loss, commonly seen with high-volume injections in the superficial plane.

The recent study by Hong et al. (63) discusses the benefits of the sub-SMAS injection plane. The authors argue that injecting firm fillers into the sub-SMAS layer achieves both volume restoration and natural lifting effects while minimizing complications and avoiding unnatural appearances. This technique increases the density of the sub-SMAS tissue, leading to a tightening effect on the SMAS layer. This tension not only supports the SMAS layer but also creates a traction effect that extends to adjacent areas, resulting in a natural lifting effect. By understanding and utilizing the distinct mechanical properties and anatomical variability of facial tissues, the sub-SMAS injection technique ensures more precise and harmonious aesthetic outcomes.

We have incorporated these recent studies to provide a more comprehensive and up-to-date discussion on facial anatomy, addressing the prevention and management of overfilled syndrome, the importance of the transverse facial septum, and the efficacy of sub-SMAS injection techniques.

Thank you for your guidance and the opportunity to improve our manuscript. We hope these revisions meet your expectations.

Sincerely,

Round 3

Reviewer 2 Report

Comments and Suggestions for Authors

Paper can be published after clearing the copyright and ethical issues - at the discretion of the Editorial Office